

# IPI score as a new prognostic index in extensive stage small cell lung cancer

Ahmet Burak Ağaoğlu, Ferhat Ekinci, Mustafa Şahbazlar and Atike Pınar Erdoğan

Division of Medical Oncology, Faculty of Medicine, Celal Bayar University, Manisa, Turkey

## ABSTRACT

**Background**. Personalized prognostic assessment in extensive-stage small cell lung cancer (ES-SCLC) necessitates a comprehensive understanding of systemic inflammatory markers and their impact on survival outcomes. This study aimed to evaluate the prognostic significance of a novel Inflammatory Prognostic Index (IPI) score, derived from four inflammation-related biochemical markers—albumin, C-reactive protein (CRP), neutrophils, and lymphocytes—in patients with ES-SCLC.

**Methods**. Patients diagnosed with ES-SCLC were eligible if adequate clinical, pathological, and follow-up data were available. The IPI score was derived using the formula: C-reactive protein × neutrophil-to-lymphocyte ratio (NLR)/serum albumin. The threshold value for the IPI score was identified using receiver operating characteristic (ROC) curve analysis within the cohort and was applied in an exploratory manner. Based on the predefined cut-off, patients were stratified into low- and high-IPI groups. The log-rank test was used to compare survival times, while Kaplan–Meier curves and Cox regression analyses assessed variables associated with long-term survival. Overall survival (OS) served as the primary endpoint, and progression-free survival (PFS) was evaluated as a secondary endpoint.

**Results**. Patients with a high IPI score had a mean OS of 9 months (95% CI [4.8–13.2]), while those with a low IPI score had a mean OS of 23 months (95% CI [11.4–34.6]), a statistically significant difference ($p = 0.005$). The prognostic significance of IPI was confirmed in both univariate ($p = 0.003$) and multivariate ($p = 0.012$) analyses.

**Conclusion**. The IPI score in ES-SCLC patients was associated with prognosis, with a high IPI score indicating poorer OS. These findings should be considered hypothesis-generating and warrant validation in larger prospective cohorts.

## INTRODUCTION

Lung cancer remains among the most frequently diagnosed malignancies worldwide and is the leading cause of cancer-related deaths (*Bray et al., 2024*). Lung cancer is of two principal subtypes. One of these, small cell lung cancer (SCLC), is a neuroendocrine tumor marked by rapid progression, poor differentiation, and high malignancy, with a survival rate of around 7% over five years (*Schwendenwein et al., 2021*). SCLC accounts for about 14–17% of all lung cancer cases. The remaining cases are broadly categorized as non-small cell lung cancer (NSCLC), comprising several histological subtypes and

Corresponding author
Ahmet Burak Ağaoğlu,
abagaoglu@hotmail.com

typically associated with a more favorable prognosis than SCLC. A majority of SCLC cases (around 80–85%) are identified at advanced or extensive stages, with two-year overall survival (OS) observed in only 21.7% of patients, and median OS notably shorter than in limited-stage disease (8.7 months *versus* 16.9 months) (*Oronsky et al., 2022*; *Huang et al., 2021*; *Demedts, Vermaelen & Van Meerbeeck, 2009*). Tumor-specific characteristics and stage at diagnosis are key determinants of recurrence risk, treatment efficacy, and survival in SCLC and other cancers. However, even among patients with comparable stages and histological features, clinical trajectories often vary, highlighting the need for additional prognostic markers (*Park et al., 2014*). The identification of such markers may enhance risk stratification, enable more accurate mortality prediction, and help tailor treatment and follow-up strategies.

As research into the tumor microenvironment advances, the intricate connections between immunity, inflammation, and cancer have become increasingly apparent. Inflammation plays a pivotal role in nearly all stages of tumorigenesis, from its initiation and promotion to eventual metastasis. Additionally, immune surveillance is vital for preventing or controlling tumor growth (*Grivennikov, Greten & Karin, 2010*). Over the years, various combinations of biomarkers have been evaluated in SCLC to estimate recurrence risk, predict treatment outcomes, and determine prognosis. Examples include the neutrophil-to-lymphocyte ratio (NLR), the platelet-to-lymphocyte ratio (PLR), and the Glasgow Prognostic Score, a composite index based on albumin and C-reactive protein (CRP). Recently, an inflammatory prognostic index (IPI) has been developed for NSCLC, incorporating CRP levels, NLR, and serum albumin. Studies have shown that elevated IPI values are associated with poorer survival in patients with NSCLC (*Dirican et al., 2016*).

This study focused on evaluating the association between IPI scores and survival outcomes, including progression-free survival (PFS) and OS, in extensive-stage small cell lung cancer patients (ES-SCLC).

## METHODS

### Patient selection criteria

This retrospective study reviewed data from patients diagnosed with SCLC, confirmed to be in the extensive stage, and followed up at Manisa Celal Bayar University, Turkey, between January 2012 and November 2024. Inclusion criteria were as follows: (I) patients aged 18 years or older with histologically confirmed ES-SCLC; (II) availability of baseline blood tests and complete clinical, pathological, and follow-up data; (III) patients receiving first-line treatment, including chemotherapy and/or immunotherapy. Exclusion criteria: (I) patients with a second primary malignancy, benign/malignant hematologic diseases, or acute infections; (II) those with chronic inflammatory or autoimmune diseases that could interfere with inflammatory marker analysis; (III) patients without adequate laboratory and clinical follow-up data.

A total of 94 patients met the inclusion criteria. However, IPI values could not be calculated for 14 patients due to missing laboratory parameters. Thus, 80 patients were included in IPI-based subgroup and PFS analyses, whereas OS analyses were conducted

in the entire cohort ($n = 94$). Tumor staging followed the guidelines of the 8th edition of the American Joint Committee on Cancer (AJCC) TNM staging system. This study was approved by the Ethics Committee of Manisa Celal Bayar University, Faculty of Medicine (decision number 20.478.486/2686, October 15, 2024), and written informed consent was obtained from all patients or their legal representatives.

## Data collection

Clinical and pathological data, including age, sex, Eastern Cooperative Oncology Group-performance status (ECOG-PS), treatment regimens, and metastatic sites, were retrieved from electronic medical records. Baseline laboratory parameters, including CRP (mg/L), neutrophil ($\times 10^9$/L; N), lymphocyte ($\times 10^9$/L; LYM), and albumin (g/dL; ALB) were recorded within one week before diagnosis. We selected pre-diagnostic laboratory values to ensure untreated baseline measurements, avoiding potential influences from biopsy-related complications, medical treatments, or radiotherapy. At the time of histological confirmation, only hemogram and coagulation tests are routinely available. Inflammatory markers, including NLR and IPI, were calculated using the formula IPI = (CRP $\times$ NLR)/ALB (*Dirican et al., 2016*). The primary study endpoints were OS and PFS, while tumor response was systematically assessed according to RECIST version 1.1.

## Follow-up and outcomes

Patients were followed up through outpatient visits, inpatient records, and telephone consultations. OS was measured from diagnosis to death or last follow-up; PFS was defined from the initiation of first-line therapy to radiographic or clinical progression or death from any cause, whichever occurred first. A threshold for the IPI score was determined by analyzing OS and PFS through receiver operating characteristic (ROC) curve analysis, which identified the optimal cut-off value, along with sensitivity and specificity, to assess its prognostic performance. The IPI was analyzed both as a continuous variable and, secondarily, dichotomized at 1.25 (ROC-derived cut-off). An optimal cut-off value of 1.25 for overall survival was identified and used to stratify patients into low- and high-IPI groups.

## Statistical analysis

Associations between clinicopathological characteristics and the IPI score were analyzed through the Chi-square test to determine any significant correlations. The relationship between OS and IPI scores was examined using Kaplan–Meier survival curves, with statistical significance assessed through Cox regression analysis. Patients were stratified into low- and high-IPI groups, with survival differences assessed by the log-rank test. Normally distributed data were analyzed using Student's $t$-test, and non-normally distributed data using the Mann–Whitney U-test. Univariate and multivariate Cox regression models were applied to calculate hazard ratios (HRs) with 95% confidence intervals (CIs) and to identify variables associated with survival outcomes. In multivariate models, only the composite IPI score was included; its individual components (CRP, NLR, albumin) were not entered simultaneously, thereby avoiding collinearity. To minimize overfitting, variables with $p < 0.10$ in univariate analysis were prioritized for inclusion in the multivariate model, and
clinically relevant covariates were also considered irrespective of univariate significance. Post-baseline variables (such as number of treatment lines) were treated as exploratory. All statistical tests were two-tailed, and $p < 0.05$ was considered statistically significant. Analyses were performed using SPSS software, version 15.0 (SPSS Inc., Chicago, IL, USA).

## RESULTS

### Patient characteristics

A retrospective evaluation was conducted on 94 patients diagnosed with ES-SCLC. Of these, 90.4% were male ($n = 85$) and 9.6% were female ($n = 9$). The mean age was 62.6 years (range, 43–80). Bone was the most common metastatic site, present in 59.6% of patients ($n = 56$). Throughout the follow-up period, 82% ($n = 77$) of patients had died, while 18% ($n = 17$) remained alive. With respect to treatment, eight patients did not receive systemic therapy, while six patients were treated with chemo-immunotherapy (atezolizumab or durvalumab plus platinum–etoposide), and two patients received other chemotherapy regimens. The remaining patients received standard platinum–etoposide doublet chemotherapy. The overall response rate (ORR) to first-line therapy was 64.1%, including complete response in eight patients (8.5%), partial response in 43 (45.7%), stable disease in 10 (10.6%), and progressive disease in 22 patients (23.4%). Radiotherapy was administered to 17 patients (including prophylactic cranial irradiation and thoracic RT). Second-line treatment was given to 37 patients (39.4%), of whom 24 received the temozolomide–irinotecan combination, nine received irinotecan monotherapy, and four received other chemotherapy regimens.

### Association between IPI score and clinical features

Based on the established IPI cut-off, patients were separated into two groups: those with an IPI $\leq 1.25$ (Low IPI) and those with an IPI $> 1.25$ (High IPI). Median follow-up was significantly longer in the Low IPI group ($p = 0.004$). Hypertension was more prevalent in the High IPI group (45.7%) compared with the Low IPI group (23.5%) ($p = 0.042$). Gender, age, ECOG-PS, and body mass index (BMI) did not differ significantly between the groups. The associations between IPI score and clinical characteristics are summarized in Table 1.

### Survival outcomes

The median follow-up duration for the entire cohort was 11 months (range, 0–76). Among surviving patients, the median follow-up was 25 months, whereas for deceased patients it was 9 months. Receiver operating characteristic (ROC) curve analysis identified 1.25 as the optimal IPI cut-off for predicting overall survival. The area under the curve (AUC) was 0.617 (95% CI [0.502–0.724]), with both sensitivity and specificity of 62.5%. The 1-year OS rate was 46.3%, with significantly poorer survival in the high IPI group compared to the low IPI group (41.8% *vs.* 60.7%, $p < 0.05$). Likewise, the 3-year OS rate was 15.9%, and those with a high IPI had notably lower 3-year OS compared to individuals with a low IPI (8.8% *vs.* 32.3%, $p < 0.05$). Furthermore, individuals in the high IPI group exhibited a median OS of 9 months (95% CI [4.8–13.2]), while those with a low IPI score had a

**Table 1  Assessment of categorical variables in relation to the IPI cut-off point.**

| Characteristic | Total (*n* = 80) | ≤1.25 (*n* = 34) | >1.25 (*n* = 46) | *p*-value |
|---|---|---|---|---|
| Age, years (mean ± SD) | 62.6 ± 8.2 | 61.1 ± 8.5 | 63.3 ± 7.7 | 0.219 |
| Sex, *n* (%) | | | | |
| Female | 9 (11.2%) | 6 (17.6%) | 3 (6.5%) | 0.159 |
| Male | 71 (88.8%) | 28 (82.40%) | 43 (93.5%) | |
| Smoking, pack-years, *n* (%) | | | | |
| Active smoker | 50 (62.5%) | 21 (61.7%) | 29 (63.0%) | |
| Ex-smoker | 29 (36.2%) | 12 (35.2%) | 17 (27.1%) | 1.000 |
| Non-smoker | 1 (1.2%) | 1 (2.9%) | 0 (0.0%) | |
| Smoking, pack-years (median, min–max) | 43 (0–140) | 42.5 (0–135) | 45 (10–140) | 0.795 |
| BMI (mean ± SD) | 25.7 ± 4.4 | 25.4 ± 4.4 | 25.8 ± 4.5 | 0.731 |
| Comorbid disease, *n* (%) | | | | |
| HT | 29 (34.0%) | 8 (23.5%) | 21 (45.7%) | 0.042 |
| DM | 20 (24.5%) | 9 (26.5%) | 11 (23.9%) | 0.794 |
| CAD | 12 (16.0%) | 4 (11.8%) | 8 (17.4%) | 0.486 |
| CRF | 3 (3.2%) | 1 (2.9%) | 2 (4.3%) | 1.000 |
| ECOG PS, *n* (%) | | | | |
| 0–1 | 68 (85%) | 30 (88.2%) | 38 (82.6%) | 0.486 |
| ≥2 | 12 (15.0%) | 4 (11.8%) | 8 (17.4%) | |
| Metastasis status, *n* (%) | | | | |
| Liver metastasis | 33 (41.5%) | 12 (35.3%) | 21 (45.7%) | 0.352 |
| Lung metastasis | 42 (59.6%) | 18 (52.9%) | 24 (52.2%) | 0.946 |
| Bone metastasis | 52 (59.6%) | 22 (64.7%) | 30 (65.2%) | 0.962 |
| Brain metastasis | 30 (39.8%) | 13 (39.4%) | 17 (37.0%) | 0.826 |
| Other metastasis | 29 (37.2%) | 10 (29.4%) | 19 (41.3%) | 0.274 |
| Number of treatment lines, *n* (%) | | | | |
| 0 | 3 (3.7%) | 1 (2.9%) | 2 (4.3%) | |
| 1 | 44 (55.0%) | 15 (44.1%) | 29 (63.0%) | |
| 2 | 19 (23.7%) | 9 (26.5%) | 10 (21.7%) | 0.414 |
| 3 | 10 (12.5%) | 6 (17.6%) | 4 (8.7%) | |
| 4 | 1 (1.2%) | 1 (2.9%) | 0 (0.0%) | |
| 5 | 3 (3.7%) | 2 (5.9%) | 1 (2.2%) | |
| Follow-up time, months (median, min–max) | 11 (0–76) | 23 (0–55) | 9 (0–76) | **0.004** |

**Notes.**

In this table, $n = 80$ indicates patients with available IPI values used for cut-off–based analyses. The results are presented as mean ± SD or count (%), with $p \leq 0.05$ considered significant. Independent *t*-test or Mann–Whitney U test was used depending on the data distribution, while the Chi-square test was applied for categorical variables to evaluate the association with the IPI score.

Abbreviations: CAD, Coronary artery disease; DM, Diabetes mellitus; HT, Hypertension; CRF, Chronic renal failure.

Bold values indicate statistically significant results ($p \leq 0.05$).

median OS of 23 months (95% CI [11.4–34.6]), highlighting the considerable prognostic influence of the IPI score on survival, as illustrated in Fig. 1. Median PFS was 7 months (95% CI [5.9–8.1]) in the low-IPI group and 6 months (95% CI [4.4–7.6]) in the high-IPI group; however, this difference did not reach statistical significance (log-rank $p = 0.542$).

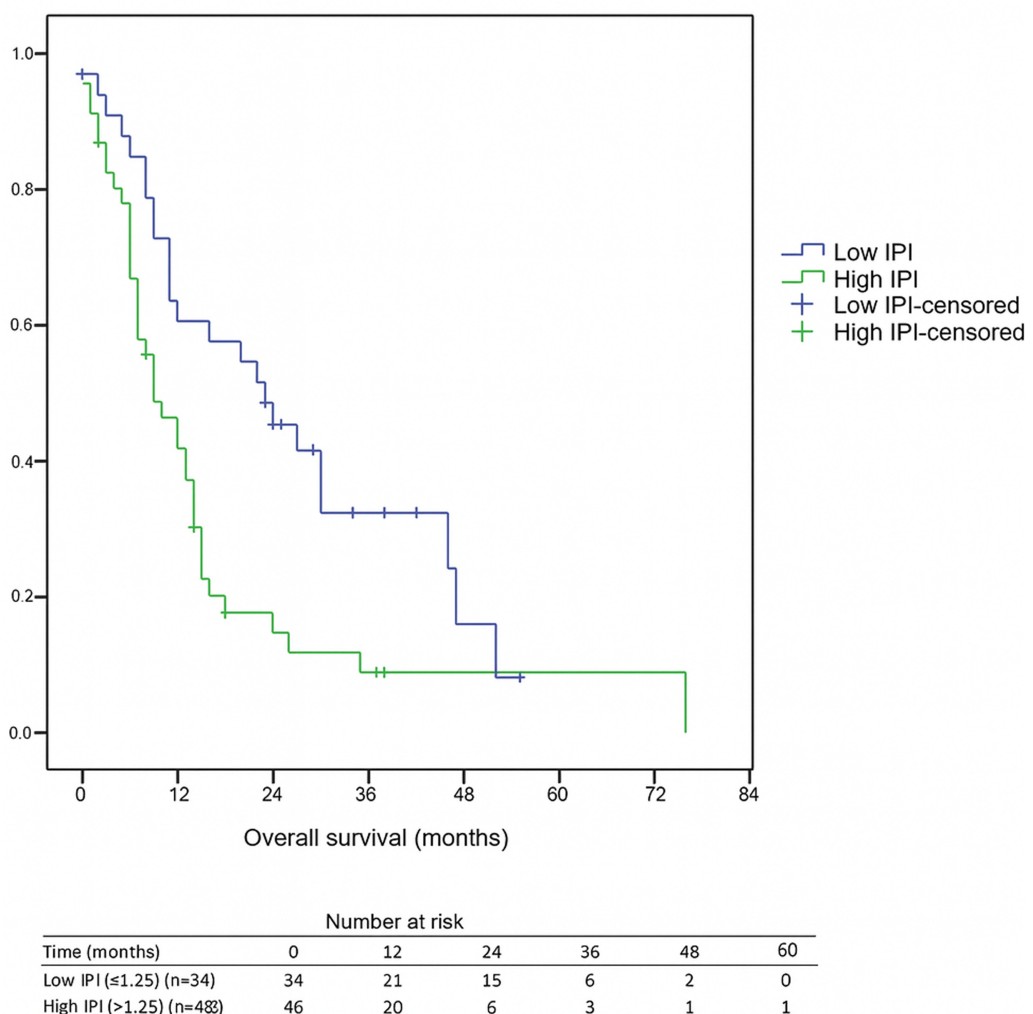

| Number at risk | | | | | |
|---|---|---|---|---|---|
| Time (months) | 0 | 12 | 24 | 36 | 48 | 60 |
| Low IPI (≤1.25) (n=34) | 34 | 21 | 15 | 6 | 2 | 0 |
| High IPI (>1.25) (n=48) | 46 | 20 | 6 | 3 | 1 | 1 |

**Figure 1** **Kaplan–Meier overall survival by IPI group.** Numbers at risk are shown at 0, 12, 24, 36, 48, and 60 months below the *x*-axis. Median OS: Low IPI = 23 months (95% CI [11.4–34.6]) *vs.* High IPI = 9 months (95% CI [4.8–13.2]); log-rank $p = 0.005$. Abbreviations: IPI, inflammatory prognostic index; OS, overall survival.

## Cox regression analysis findings

A Cox regression analysis was conducted to thoroughly assess both PFS and OS using univariate and multivariate models. PFS was significantly associated with brain and bone metastases, chronic renal failure, and administration of second-line therapy, as shown in Table 2. However, the IPI score did not demonstrate independent prognostic value for PFS. Conversely, survival analysis revealed significant associations with liver and bone metastases, BMI, the number of treatment lines, high IPI, and type 2 diabetes, as seen in Table 3. In univariate analysis, an IPI > 1.25 was associated with a 1.029-fold higher risk of death, while in multivariate analysis the risk was 1.081-fold, establishing IPI as an independent prognostic factor for survival.

**Table 2  Cox regression analysis results of the PFS.**

| | Univariate | | Multivariate | |
|---|---|---|---|---|
| | HR (95% CI) | *p*-value | HR (95% CI) | *p*-value |
| Female | | | | |
| Male | 1.497 (0.717–3.127) | 0.283 | 3.249 (0.992–10.648) | 0.052 |
| Age | 1.023 (0.993–1.053) | 0.142 | 1.026 (0.979–1.076) | 0.287 |
| Non smoker, ex smoker | | | | |
| Active smoker | 0.947 (0.990–1.490) | 0.814 | 0.873 (0.452–1.685) | 0.685 |
| Pack-year smoking | 0.998 (0.990–1.006) | 0.635 | 0.993 (0.982–1.004) | 0.206 |
| BMI | 1.005 (0.961–1.052) | 0.820 | 1.046 (0.953–1.148) | 0.341 |
| Comorbid disease | | | | |
| HT | 1.127 (0.710–1.789) | 0.612 | 1.261 (0.576–2.762) | 0.562 |
| DM | 1.070 (0.650–1.762) | 0.790 | 1.406 (0.581–3.401) | 0.449 |
| CAD | 0.999 (0.561–1.780) | 0.997 | 0.499 (0.170–1.462) | 0.205 |
| CRF | 1.155 (0.363–3.680) | 0.807 | 4.762 (1.121–20.236) | **0.034** |
| ECOG PS | 0.949 (0.522–1.726) | 0.864 | 0.745 (0.264–2.102) | 0.578 |
| Metastasis status | | | | |
| Liver metastasis | 1.406 (0.900–2.195) | 0.134 | 0.793 (0.367–1.716) | 0.556 |
| Lung metastasis | 0.932 (0.599–1.451) | 0.756 | 1.017 (0.534–1.939) | 0.959 |
| Bone metastasis | 1.487 (0.938–2.358) | 0.092 | 2.305 (1.145–4.640) | 0.019 |
| Brain metastasis | 0.770 (0.496–1.197) | 0.246 | 0.352 (0.178–0.697) | **0.003** |
| Other metastasis | 1.127 (0.716–1.775) | 0.605 | 0.837 (0.421–1.662) | 0.610 |
| Number of treatment lines | | | | |
| 0–1 | Ref | | Ref | |
| 2 | 0.568 (0.330–0.979) | 0.042 | 0.203 (0.089–0.463) | **<0.001** |
| 3 | 0.567 (0.303–1.059) | 0.075 | 0.390 (0.151–1.005) | 0.051 |
| 4 | 0.493 (0.119–2.049) | 0.331 | 0.572 (0.051–6.477) | 0.652 |
| 5 | 0.579 (0.178–1.183) | 0.364 | 1.606 (0.329–7.827) | 0.558 |
| IPI | 1.009 (0.990–1.030) | 0.349 | 0.996 (0.945–1.051) | 0.051 |

**Notes.**
$n = 80$ patients were included in the PFS analysis. IPI score and PFS correlations were analyzed through Kaplan–Meier methodology and assessed using log-rank tests, with a *p*-value of $\leq 0.05$ considered statistically significant.
Abbreviations: HR, Hazard ratio; CAD, Coronary artery disease; HT, Hypertension; DM, Diabetes mellitus; CRF, Chronic renal failure.
Bold values indicate statistically significant results ($p \leq 0.05$).

# DISCUSSION

To our knowledge, this is the first study to assess the prognostic relevance of IPI in ES-SCLC patients. Our results demonstrate that IPI was associated with OS, suggesting a potential prognostic role.

Systemic inflammatory response is widely recognized as a biomarker of host-related factors that impact cancer prognosis (*Grivennikov, Greten & Karin, 2010*). Elevated CRP levels typically indicate systemic inflammation, often driven by tumor progression. A recent Danish study demonstrated a correlation between higher CRP levels and poor survival outcomes (*Stensvold et al., 2021*). Moreover, patients with lower serum CRP levels had a significantly longer median time to treatment failure (*Brustugun, Sprauten & Helland,*

**Table 3   Cox regression analysis results of the OS.**

| | Univariate | | Multivariate | |
|---|---|---|---|---|
| | HR (95% CI) | *p*-value | HR (95% CI) | *p*-value |
| Female | | | | |
| Male | 1.392 (0.637–3.042) | 0.407 | 1.897 (0.530–6.794) | 0.325 |
| Age | 1.025 (0.997–1.055) | 0.082 | 1.001 (0.956–1.049) | 0.953 |
| Non smoker, ex smoker | | | | |
| Active smoker | 1.310 (0.810–2.212) | 0.271 | 1.078 (0.475–2.449) | 0.857 |
| Pack-year smoking | 0.997 (0.989–1.005) | 0.508 | 0.993 (0.980–1.006) | 0.274 |
| BMI | 0.955 (0.909–1.004) | 0.073 | 0.903 (0.827–0.986) | **0.023** |
| Comorbid disease | | | | |
| HT | 0.925 (0.574–1.494) | 0.751 | 0.441 (0.188–1.033) | 0.059 |
| DM | 1.052 (0.617–1.793) | 0.852 | 2.972 (1.191–7.416) | **0.020** |
| CAD | 0.889 (0.467–1.694) | 0.721 | 1.261 (0.402–3.951) | 0.691 |
| CRF | 0.824 (0.210–3.369) | 0.787 | 3.876 (0.686–21.906) | 0.125 |
| ECOG PS | 1.230 (0.687–2.203) | 0.487 | 0.861 (0.328–2.258) | 0.761 |
| Metastasis status | | | | |
| Liver metastasis | 1.908 (1.192–3.053) | **0.007** | 2.230 (1.086–4.577) | **0.029** |
| Lung metastasis | 0.714 (0.447–1.141) | 0.159 | 0.675 (0.340–1.340) | 0.261 |
| Bone metastasis | 1.169 (0.739–1.850) | 0.504 | 2.408 (1.157–5.013) | **0.019** |
| Brain metastasis | 0.782 (0.448–1.252) | 0.306 | 0.506 (0.254–1.008) | 0.053 |
| Other metastasis | 1.431 (0.899–2.279) | 0.131 | 1.226 (0.588–2.556) | 0.587 |
| IPI | 1.029 (1.010–1.049) | **0.003** | 1.081 (1.017–1.150) | **0.012** |
| Number of treatment lines | | | | |
| 0–1 | Ref | | Ref | |
| 2 | 0.324 (0.172–0.612) | **0.001** | 0.186 (0.072–0.481) | **0.001** |
| 3 | 0.258 (0.121–0.551) | **<0.001** | 0.194 (0.055–0.682) | **0.011** |
| 4 | 0.402 (0.097–1.669) | 0.209 | 1.130 (0.089–14.361) | 0.925 |
| 5 | 0.302 (0.073–1.258) | 0.100 | 0.992 (0.129–7.634) | 0.994 |

**Notes.**
$n = 94$ patients were included in the OS analysis. IPI score and OS correlations were evaluated using Kaplan–Meier curves with log-rank statistics, with a *p*-value of $\leq 0.05$ considered statistically significant.
Abbreviations: HR, Hazard ratio; CAD, Coronary artery disease; HT, Hypertension; DM, Diabetes mellitus; CRF, Chronic renal failure.
Bold values indicate statistically significant results ($p \leq 0.05$).

*2016*). These findings suggest that elevated CRP levels may be predictive of worse survival outcomes.

Neutrophils release various enzymes that create a microenvironment favorable for angiogenesis and invasion. These mechanisms may also contribute to treatment resistance. Conversely, lymphocytes are essential for anti-tumor immunity, contributing to the inhibition of tumor growth and metastasis (*Houghton et al., 2010*; *Mantovani et al., 2008*). An elevated NLR signifies an imbalance, characterized by increased neutrophil counts and decreased lymphocyte levels. Accordingly, increased neutrophil levels can contribute to tumor progression, while decreased lymphocyte counts compromise anti-tumor immune responses. A systematic review on NLR in SCLC underlined its association with OS,

highlighting its role as a prognostic indicator (*Winther-Larsen, Aggerholm-Pedersen & Sandfeld-Paulsen, 2021*).

Serum albumin is a commonly used marker of disease severity, an established prognostic indicator in cancer, and a tool for evaluating the efficacy of immune checkpoint inhibitors (*Kuang, Miao & Zhang, 2024*). Hypoalbuminemia, often resulting from systemic inflammation, reflects malnutrition or cachexia associated with tumor progression (*Kuang, Miao & Zhang, 2024*).

When combined, CRP, NLR, and serum albumin form a more robust scoring system to predict SCLC prognosis. A high IPI, characterized by elevated CRP and neutrophil counts together with reduced lymphocytes and serum albumin, reflects an intense inflammatory state and a weakened immune response. This imbalance may facilitate cancer invasion and metastasis, ultimately leading to poor survival outcomes.

In our study, IPI showed no correlation with age, gender, smoking status, BMI, ECOG performance status, metastatic sites, or other comorbidities, suggesting that it is not directly associated with disease burden or severity in SCLC. Nevertheless, univariate and multivariate analyses showed that IPI was associated with OS, highlighting its potential prognostic value, though these findings should be regarded as exploratory. It is noteworthy that some variables not significant in univariate analysis became significant in the multivariate model. This likely reflects the effect of confounding, where adjustment for other covariates allowed the independent prognostic effect to emerge, a phenomenon commonly observed in multivariate survival analyses. Although the observed HR was modest, this may be related to the limited sample size, treatment heterogeneity, and the aggressive nature of ES-SCLC, where multiple strong prognostic factors operate simultaneously. Even modest independent prognostic effects can provide value for clinical risk stratification, and IPI may complement existing clinical variables in guiding individualized management strategies.

Compared to other systemic inflammation scores such as NLR, PLR, GPS, and PNI, which rely on two parameters (*Winther-Larsen, Aggerholm-Pedersen & Sandfeld-Paulsen, 2021*; *Bennett, 2020*), IPI incorporates four (*Dirican et al., 2016*). IPI may serve as a more comprehensive marker, capturing the equilibrium between the host's inflammatory and immune responses better than other systemic inflammation indices.

Among inflammation-based scores, NLR and PLR are the most commonly studied in SCLC. In a meta-analysis of 16 studies, high NLR was associated with a 39% increased risk of death (HR = 1.39, 95% CI [1.22–1.56]) (*Winther-Larsen, Aggerholm-Pedersen & Sandfeld-Paulsen, 2021*). In the same meta-analysis, however, PLR's correlation with OS was not confirmed (HR = 1.20, 95% CI [0.96–1.52]) (*Winther-Larsen, Aggerholm-Pedersen & Sandfeld-Paulsen, 2021*). One explanation is that PLR incorporates only one general inflammation marker, lymphocytes. These findings highlight the need to refine current inflammation-based scoring systems. Another meta-analysis involving 2,831 SCLC patients found that a pretreatment high GPS was significantly associated with poorer OS (HR = 1.90, 95% CI [1.36–2.63]) (*Xie, Li & Hu, 2023*). Similarly, in a meta-analysis of nine studies including 4,164 SCLC patients, low PNI status was closely linked to decreased OS (HR =
1.42, 95% CI [1.24–1.64]) (*Jiang et al., 2020*). Monitoring PNI and improving immune and nutritional status may help enhance the prognosis of SCLC patients.

The IPI score has been tested as a prognostic tool in multiple cancers, including its association with survival and treatment response. Dirican et al. first demonstrated the utility of IPI in predicting outcomes in NSCLC (*Dirican et al., 2016*). Their findings revealed a significantly shorter median OS in the high IPI group compared to the low IPI group (8.0 *vs.* 34.0 months; HR: 3.5; *p* < 0.001), highlighting its role as a distinctive prognostic marker for OS in NSCLC. Moreover, the IPI score has also been recognized as a notable prognostic marker for OS in patients with colorectal, renal, and gastric carcinomas (*Ekinci et al., 2022*; *Erdoğan et al., 2021*; *Ekinci et al., 2021*).

IPI offers several advantages: it is widely available, cost-effective, non-invasive, and requires no specialized equipment, making it a practical tool for prognostic assessment in ES-SCLC. Further large-scale, multicenter studies across diverse populations are warranted to validate these findings and refine recommendations.

It is important to acknowledge that treatment regimens in our cohort were heterogeneous. While most patients received standard platinum–etoposide chemotherapy, a small subset received chemo-immunotherapy, alternative regimens, or no systemic treatment at all. Radiotherapy was administered to selected patients, and various second-line therapies were employed, including temozolomide–irinotecan combinations and irinotecan monotherapy. This heterogeneity reflects real-world clinical practice but may confound prognostic analyses, as differences in treatment exposure could influence outcomes independently of IPI. Importantly, the prognostic significance of IPI for overall survival persisted in multivariate analysis after adjustment for treatment lines. The observed overall response rate of 64% in our cohort falls within the expected range reported for platinum–etoposide and chemo-immunotherapy regimens in ES-SCLC (*Oronsky et al., 2022*; *Huang et al., 2021*; *Demedts, Vermaelen & Van Meerbeeck, 2009*). Likewise, the proportion of patients who were able to receive second-line therapy (39%) is consistent with real-world series, in which only 20–40% of patients typically proceed beyond first-line treatment (*Jiang et al., 2020*; *Stensvold et al., 2021*). Furthermore, the second-line regimens administered in our study, such as irinotecan-based or temozolomide combinations, are consistent with NCCN-recommended options for relapsed SCLC (*NCCN, 2024*).

This ES-SCLC research has limitations for future interpretations. Firstly, its retrospective, single-center design inherently restricts the generalizability of the findings, as it is based on a relatively small sample size. Secondly, the IPI cutoff value identified in this study was derived from ROC analysis within our cohort and has not been externally validated. As this is the first study assessing IPI in ES-SCLC, the optimal threshold may differ in larger or independent datasets, and our findings should be regarded as hypothesis-generating. Additionally, we did not perform further discrimination or calibration analyses, which limits the evaluation of predictive performance. Although the study spanned a long period, treatment standards remained largely unchanged, aside from the recent introduction of chemo-immunotherapy. Despite stable treatment standards, the extended timeframe may have introduced heterogeneity in treatment and supportive care. Moreover, we did not stratify patients according to treatment era, which may have influenced outcomes due

to evolving standards of care. Finally, the predominance of male patients represents a potential source of gender bias that could influence the interpretation of our findings, and IPI values were unavailable in 14 patients due to missing laboratory data. As a result, IPI-based subgroup and PFS analyses included 80 patients, while OS analyses covered all 94.

To validate these findings, future research should be conducted in multicenter, prospective cohorts including larger and more diverse populations. Additionally, efforts to standardize the calculation and interpretation of the IPI score across diverse settings are essential to establish its clinical utility. Addressing potential gender differences in prognosis and treatment response should also be a focus of subsequent studies. Although our study evaluated IPI only at baseline, future prospective studies should investigate whether dynamic changes in IPI during and after treatment may provide additional prognostic information.

## CONCLUSIONS

Our findings indicate that the IPI score is a significant prognostic indicator in ES-SCLC. Patients with a high IPI score were reported to have significantly worse OS. To validate these findings, further studies with larger patient cohorts and prospective designs are necessary.

### Funding
The authors received no funding for this work.

### Competing Interests
The authors declare there are no competing interests.

### Author Contributions
- Ahmet Burak Ağaoğlu conceived and designed the experiments, analyzed the data, prepared figures and/or tables, authored or reviewed drafts of the article, and approved the final draft.
- Ferhat Ekinci performed the experiments, analyzed the data, prepared figures and/or tables, authored or reviewed drafts of the article, and approved the final draft.
- Mustafa Şahbazlar performed the experiments, prepared figures and/or tables, authored or reviewed drafts of the article, and approved the final draft.
- Atike Pınar Erdoğan conceived and designed the experiments, prepared figures and/or tables, authored or reviewed drafts of the article, and approved the final draft.

### Human Ethics
The following information was supplied relating to ethical approvals (*i.e.*, approving body and any reference numbers):

The Ethics Committee of Manisa Celal Bayar University, Faculty of Medicine, approved this study on October 15, 2024 (decision number 20.478.486/2686).

## Data Availability

The raw data is available in the Supplemental Files.

## Supplemental Information

Supplemental information for this article can be found online at http://dx.doi.org/10.7717/peerj.20343#supplemental-information.

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
