# Peer review of "IPI score as a new prognostic index in extensive stage small cell lung cancer"

_PeerJ, doi:10.7717/peerj.20343_

## Round 0.1 · original submission · Major Revisions

· Academic Editor

Major Revisions

**Language Note:** When preparing your next revision, please ensure that your manuscript is reviewed either by a colleague who is proficient in English and familiar with the subject matter, or by a professional editing service. PeerJ offers language editing services; if you are interested, you may contact us at [email protected] for pricing details. Kindly include your manuscript number and title in your inquiry. – PeerJ Staff

Reviewer 1 ·

Basic reporting

The manuscript is written in generally clear English, but several key data points are missing that undermine transparency and completeness. The authors do not report the median duration of follow-up, making it impossible to assess the maturity of survival data or detect potential censoring bias between high‐ and low‐IPI groups. Likewise, the overall response rate to first‐line therapy is not provided, even though this metric is essential for determining whether a high IPI merely reflects poorer initial tumor response. Details on systemic treatment regimens are entirely absent: there is no breakdown of how many patients received single‐agent chemotherapy versus combination chemotherapy doublets or chemo‐immunotherapy (and which immune‐checkpoint inhibitor was used), nor is there any mention of whether prophylactic cranial irradiation or thoracic radiotherapy was administered. Without these variables, follow‐up duration, ORR, specific first‐line regimens, and radiotherapy details, the reporting fails to present a complete clinical picture and invites confounding by treatment differences.

Experimental design

The study’s small, single‐center, retrospective design (with a predominantly male cohort) limits generalizability and raises concerns about statistical power. Some hazard ratios, for example, IPI versus PFS, hover near significance, suggesting possible type I or II errors that a larger sample might resolve. The IPI cutoff of 1.25 was derived via ROC analysis on the same dataset used for outcome assessment; no internal validation (such as bootstrapping) or external cohort testing was performed, risking overfitting and undermining applicability to other populations. Important prognostic biomarkers, such as LDH, neuron‐specific enolase, and quantitative tumor burden (e.g., number of metastatic sites), were not included, and the manuscript does not explain how missing data were handled. The second‐line treatment categories are not well defined (we do not know which regimens patients received beyond first line), and there is no discussion of paraneoplastic syndromes, which are common in extensive‐stage SCLC and can markedly influence systemic inflammation. Finally, the statistical models do not address multicollinearity among inflammatory markers (CRP, NLR, albumin), nor do they report discrimination metrics or model calibration, leaving the clinical utility of IPI unclear.

Validity of the findings

Taken together, these methodological shortcomings compromise confidence in the conclusions. Without a clear description of follow‐up duration, ORR, treatment regimens (chemo vs. chemo-IO vs. radiotherapy), and key biomarkers, the prognostic model is likely confounded by unmeasured variables. The lack of internal or external validation of the IPI cutoff raises the possibility that results are specific to this cohort only. Omitting LDH, NSE, tumor burden, and paraneoplastic syndrome data further undermines the assertion that IPI is independently prognostic. Because some hazard ratios barely reach statistical significance and the model’s discrimination and calibration are unreported, we cannot determine whether IPI meaningfully stratifies patient risk in clinical practice. Before firm conclusions can be drawn, the authors should rerun analyses stratified by treatment type, incorporate additional clinical variables, validate the cutoff, and transparently report follow-up, response rates, and missing data handling.

·

Basic reporting

1.1 The manuscript is generally well written and clearly structured. However, several typographical and stylistic issues should be addressed. For instance, “seperated” should be corrected to “separated,” and some complex sentences would benefit from rephrasing for clarity.

1.2 The introduction effectively provides the necessary background, highlighting the need for novel prognostic markers in extensive-stage small cell lung cancer (ES-SCLC). Citations are up-to-date and relevant.

1.3 Figures and tables are appropriate, clearly labeled, and informative. I recommend that the authors include the ROC analysis details (AUC, sensitivity, specificity) in the main text, not only in the figure.

Experimental design

2.1 This is a retrospective, single-center study with clearly defined inclusion and exclusion criteria. The IPI score formula (CRP × NLR / albumin) is clearly described and appears reproducible.

2.2 It would be helpful to specify whether laboratory parameters (CRP, NLR, albumin) were collected under standardized conditions. These values can be influenced by transient infections or acute inflammation, which could confound the analysis.

2.3 The ROC analysis is appropriate for identifying the IPI threshold, but the authors should report the AUC value along with sensitivity and specificity in the results section to enhance methodological transparency.

Validity of the findings

3.1 The study provides evidence that a high IPI score is independently associated with worse overall survival in ES-SCLC patients. However, no significant association was found with progression-free survival (PFS).

3.2 The hazard ratio (HR) for IPI in multivariate analysis (HR = 1.081) is statistically significant but modest in magnitude. The authors should discuss the clinical relevance of this effect more critically.

3.3 The lack of external validation is a major limitation. This should be emphasized more strongly in the discussion, and the authors should acknowledge the need for prospective, multicenter studies to confirm their findings.

Additional comments

4.1 The study population is heavily skewed toward males (90%), which may introduce gender bias. The authors should address how this may affect generalizability.

4.2 The authors should elaborate on how the IPI score might be used in clinical practice. For instance, could it assist in risk stratification, guide treatment intensity, or inform patient monitoring strategies?

4.3 Although beyond the scope of this retrospective study, the authors may wish to comment on whether changes in the IPI score over time could have prognostic value (e.g., pre- vs. post-treatment), as this could be explored in future research.

Reviewer 3 ·

Basic reporting

The study is interesting despite being retrospective and lacking novelty, as numerous existing studies have already addressed inflammatory markers in lung cancer, including SCLC. While it adds some information specific to SCLC, the manuscript falls short in several critical areas and cannot be recommended for publication in its current form.

1. The study is retrospective, and ethics committee approval was obtained in 2024. The authors claim that written informed consent was obtained from every patient. This requires clarification—how was informed consent obtained retrospectively, and what documentation supports this claim?

2. The rationale for conducting a blood test one week prior to diagnosis is unclear. The authors must explain the clinical or methodological justification for this timing.

3. It is not evident how the timing and consistency of the blood tests were assured. The manuscript must describe the procedures used to ensure standardization and reliability of the data collection.

4. The statistical validity is questionable. Factors that were not significant in univariate analysis appear significant in multivariate analysis. The authors must explain this discrepancy and provide a robust justification for the statistical approach used.

5. The treatment protocols appear to be heterogeneous. The manuscript must detail the treatment regimens and account for variability in therapeutic approaches across the study period.

6. The study spans a long timeframe, during which both treatment standards and blood test methodologies may have changed. This introduces significant heterogeneity that must be addressed or controlled for in the analysis.

7. The analysis does not account for temporal factors, such as differences between recent and older cases. This is a critical omission and must be corrected to ensure the validity of the findings.

Experimental design

-

Validity of the findings

-

---

## Round 0.2 · Major Revisions

· Academic Editor

Major Revisions

Thank you for submitting your manuscript. Reviewer 1 has provided detailed and constructive comments, while Reviewer 2 recommends acceptance. However, significant issues must be addressed before further consideration:

1) Data consistency and reporting: Reconcile discrepancies in sample size (n=94 vs. 80 in Table 1). Standardize tables and figures (remove corrupted symbols, clarify p-values, add numbers at risk to KM plots, specify biomarker units/timing).

2) Study design and analysis:
- Clarify accrual period handling; provide sensitivity analyses by therapeutic era.
- Avoid use of post-baseline variables in baseline models or model them appropriately as time-dependent.
- Re-specify OS/PFS definitions clearly.
- Address over-parameterisation; consider penalised/shrinkage Cox or reducing covariates.
- Justify or pre-specify cut-offs for dichotomised IPI; otherwise present it as a continuous variable.

3) Interpretation: Report medians (95% CI) instead of means, provide interpretable HRs, temper strength of associations, and expand limitations (risk of cut-off overfitting, accrual heterogeneity, lack of validation).

4) Figures: Provide time-dependent ROC and calibration plots if diagnostic performance is claimed; improve legends for clarity.

The paper addresses an important question, but the current presentation and analysis limit confidence in the findings. A thorough revision with substantial re-analysis and clearer reporting is required.

Reviewer 1 ·

Basic reporting

The manuscript requires clearer, more consistent reporting and language polishing. The stated sample size (n=94) does not reconcile with Table 1, where group totals sum to 80; please reconcile denominators across text, tables, and figures and explain missing data or exclusions per analysis, ideally with a simple flow diagram. Tables contain corrupted symbols (for example, “≥” rendered as “g”), garbled p-values, implausible hazard ratios, and mixed formats; please standardize all tables and add numbers at risk to Kaplan–Meier plots. Figure legends should precisely describe what is shown; if you present ROC analyses, include time-dependent ROC curves and calibration plots rather than generic “diagnostic performance.” Specify assay units and timing for all biomarkers (for example, CRP mg/L, albumin g/dL) and define the blood-draw window.

Experimental design

The accrual period (2012–2024) spans major therapeutic eras. Please provide era stratification or a sensitivity analysis excluding immunotherapy, and describe how radiotherapy and second-line treatments were handled analytically. Avoid post-baseline, prognosis-dependent variables in baseline models (for example, number of treatment lines, receipt of second-line therapy) or model them appropriately as time-dependent covariates. Define PFS from the start of first-line therapy. Given approximately 77 deaths, the multivariable OS model appears over-parameterized; reduce covariates a priori or apply penalized Cox/shrinkage with internal validation. If you wish to keep a dichotomized IPI, pre-specify the cut-off or justify it; otherwise, emphasize the continuous measure.

Validity of the findings

As currently presented, the strength of the conclusions is limited by modest discrimination (AUC), potential overfitting, and unaddressed confounding. Please replace mean survival times with medians (95% CI) and report hazard ratios from KM/Cox for high versus low IPI; retain a continuous IPI effect but scale it for interpretability. Conclusions should be softened from “strongly associated” to “associated” and framed as hypothesis-generating. Expand the limitations to acknowledge cut-off overfitting risk, post-baseline confounding, lack of external validation, and heterogeneity across the long accrual window.

·

Basic reporting

Please accept

Experimental design

please accept

Validity of the findings

please accept

---

## Round 0.3 · accepted · Accept

· Academic Editor

Accept

Thank you for your thorough revisions and thoughtful responses to the reviewers’ comments. The manuscript has significantly improved in clarity, structure, and scientific rigor. Based on the reviewer’s positive evaluation and the enhanced quality of the revised submission, I am pleased to inform you that your paper has been accepted for publication. Congratulations on this achievement, and thank you for choosing our journal for your work.

Reviewer 1 ·

Basic reporting

The authors have significantly improved the structure, the content and the background.

Experimental design

The original article has an original design.

Validity of the findings

After adjusting the results with major correction, the validity of the findings, although exclusively hypothesis-generating, is novel and could potentially have an impact.